# Magnetic Properties and Magnetocaloric Effect of Polycrystalline and Nano-Manganites Pr_0.65_Sr_(0.35−x)_Ca_x_MnO_3_ (x ≤ 0.3)

**DOI:** 10.3390/nano13081373

**Published:** 2023-04-14

**Authors:** Roman Atanasov, Dorin Ailenei, Rares Bortnic, Razvan Hirian, Gabriela Souca, Adam Szatmari, Lucian Barbu-Tudoran, Iosif Grigore Deac

**Affiliations:** 1Faculty of Physics, Babes-Bolyai University, Str. Kogalniceanu 1, 400084 Cluj-Napoca, Romania; atanasov.roman@ubbcluj.ro (R.A.); dorin.ailenei@stud.ubbcluj.ro (D.A.); rares.bortnic@ubbcluj.ro (R.B.); razvan.hirian@ubbcluj.ro (R.H.); gabriela.souca@ubbcluj.ro (G.S.); adam.szatmari@ubbcluj.ro (A.S.); 2National Institute for Research and Development of Isotopic and Molecular Technologies, Str. Donath 67-103, 400293 Cluj-Napoca, Romania; lucian.barbu@itim-cj.ro

**Keywords:** manganites, nanoparticle perovskites, crystallography, magnetic behavior, phase transition, critical behavior, magnetocaloric effect

## Abstract

Here we report investigations of bulk and nano-sized Pr_0.65_Sr_(0.35−x)_Ca_x_MnO_3_ compounds (x ≤ 0.3). Solid-state reaction was implemented for polycrystalline compounds and a modified sol–gel method was used for nanocrystalline compounds. X-ray diffraction disclosed diminishing cell volume with increasing Ca substitution in Pbnm space group for all samples. Optical microscopy was used for bulk surface morphology and transmission electron microscopy was utilized for nano-sized samples. Iodometric titration showed oxygen deficiency for bulk compounds and oxygen excess for nano-sized particles. Measurements of resistivity of bulk samples revealed features at temperatures associated with grain boundary condition and with ferromagnetic (FM)/paramagnetic (PM) transition. All samples exhibited negative magnetoresistivity. Magnetic critical behavior analysis suggested the polycrystalline samples are governed by a tricritical mean field model while nanocrystalline samples are governed by a mean field model. Curie temperatures values lower with increasing Ca substitution from 295 K for the parent compound to 201 K for x = 0.2. Bulk compounds exhibit high entropy change, with the highest value of 9.21 J/kgK for x = 0.2. Magnetocaloric effect and the possibility of tuning the Curie temperature by Ca substitution of Sr make the investigated bulk polycrystalline compounds promising for application in magnetic refrigeration. Nano-sized samples possess wider effective entropy change temperature (ΔT_fwhm_) and lower entropy changes of around 4 J/kgK which, however, puts in doubt their straightforward potential for applications as magnetocaloric materials.

## 1. Introduction

Organic life is possible in a certain range of temperature because chemical exchange is destroyed above and below that range [1]. Throughout history, while artists were driven by the need to stay “cool”, the engineering part of the human brain kept searching for more practical solutions [2]. It is obvious that the best method for cooling our environment is the one that does not harm it. As such, magnetocaloric materials present a viable alternative to harmful gasses [3,4].

Magnetocaloric effect was reported in 1881 [5], but only recently has its potential for everyday use been discussed. Gadolinium Gd has been a forerunner in excellent magnetocaloric effect for a while, until its alloys were found to have higher values of entropy change [3]. They are expensive and require fields of over 5 T for operational use [4], and so the search for even more effective compounds has begun.

Compounds of the type A_1−x_B_x_MnO_3_ (where A is a trivalent rare earth cation and B is a divalent alkaline earth cation [6] (pp. 1–153) have a perovskite structure. They allow for Mn^3+^-O-Mn^4+^ interaction where an electron from Mn^3+^ can hop to Mn^4+^, thus aligning the magnetic moments. Such interaction is named “double exchange” and is the main reason for their electric and magnetic properties [7] (pp. 18–21), [8] (pp. 167–293).

Recent work has shown that compounds such as La_1−x_Sr_x_MnO_3_ and Pr_1−x_Ba_x_MnO_3_ [9,10] exhibit large magnetic entropy change. Structurally, a divalent Sr^2+^ and Ba^2+^ are doped in place of trivalent La^3+^ and Pr^3+^. The strongest “double exchange” interaction results are achieved at the doping level of about x = 0.3 [8]. Further introduction of divalent elements leads to antiferromagnetic (AFM) arrangement and can cause localization of charge—the so-called charge ordered (CO) state [11]. Because of this, further substitution of trivalent or divalent ions with different size ions can change the structure, volume and the magnetic properties. 

In this work, we have prepared polycrystalline and nanocrystalline samples of Pr_0.65_Sr_0.35−x_Ca_x_MnO_3_ (x = 0.02, 0.05, 0.1, 0.2, 0.3). Magnetic properties of the parent bulk compound Pr_0.65_Sr_0.35_MnO_3_ have been reported previously in the literature [12]. Ca^2+^ ions act as the substitute for divalent Sr^2+^ ions in order to bring *T*_C_ down from 295 K of the parent compound Pr_0.65_Sr_0.35_MnO_3_ [12] in bulk and *T*_C_ = 257 K of nano-sized compounds. Systems were crystallographically and morphologically investigated by X-ray diffraction (XRD), optical microscopy and transmission electron microscopy (TEM), and Rietlveld refinement analysis of XRD. Stoichiometry [13] and structure [8,9] of the samples greatly affects properties. Preparation method can affect stoichiometry of the compounds, sometimes causing accidental vacancies in Pr^3+^ ions. Some changes in magnetic and electrical behavior, therefore, could be observed, but are unquantifiable by this experiment. Additionally, oxygenation plays an important role in final measurements. Oxygen deficiency and excess change stoichiometry of the compounds by changing Mn^3+^/Mn^4+^ ratio electrical conductivity and overall magnetization. In our work, oxygen content was investigated by chemical analysis of Iodometry. Critical behavior was analyzed by construction of modified Arrott plots (MAP), which was also confirmed by the Kouvel–Fisher (KF) method. Nanocrystalline compounds have lower maximum entropy change values but much wider effective temperature range. Electrical measurements, in the bulk samples, revealed colossal magnetoresistance behavior [11,14,15]. 

This paper is organized as follows: in Section 2 we describe the preparation methods, as well as all the characterization methods we used in structural, morphological, oxygen stoichiometric, electrical, and magnetic investigation. In Section 3 we present the results of our investigation, analysis of data, and we discuss the magnetic critical behavior, electrical and magnetic properties of our samples. Finally, we summarize our results in Section 4.

## 2. Materials and Methods

Polycrystalline samples were prepared by conventional solid-state reaction. High purity oxides of principal elements Pr_6_O_11_, SrO, MnO_2_, and carbonate CaCO_3_ were purchased from Alfa Aesar (Heysham, UK). After being mixed by hand in a mortar for 3 h, the powders were calcinated at 1100 °C for 24 h in air. Later, the powder was pressed into a pellet at 3 tons and sintered in air at 1350 °C for 30 h. 

Nanocrystalline samples were prepared by sucrose sol-gel method. Nitrates Pr(NO_3_)_3_ · 6H_2_O, Sr(NO_3_)_2_, Ca(NO_3_)_2_ · 4H_2_O, and Mn(NO_3_)_2_ · 6H_2_O were dissolved in pure water (18.2 MΩ × cm at 25 °C) for up to 1 h at 60 °C after which, 10 g of sucrose was added. This enables positive ions to attach themselves to OH hubs of the sucrose chain. After stirring for 45–60 min temperature was turn down to room temperature. Two grams of pectin was added to expand the xero-gel and mixed for further 20 min. Solutions were dried at 200 °C for 24 h or until all water evaporated. Finally, gels were burned at 1000 °C for 2 h to obtain the nano-sized particles. If the reaction does not happen, as was the case in this experiment, some small amount of Acetic acid can be added to the mix at the last stage before drying in order to bring the chains closer together. However, such action usually results in somewhat bigger particles. 

X-ray diffraction (XRD) was implemented for structural categorization of all samples. Optical microscopy was used on bulk samples for grain size determination and surface defects. Transmission electron microscopy (TEM) allowed for nanocrystalline particle size determination. Analysis of XRD data was performed by Rietveld refinement method, as well as Williamson–Hall (W-H) method.

Iodometric analytical titration was applied in order to detect deficiency or excess of oxygen in the samples. A small amount of a sample was placed in a closed vessel containing hydrochloric acid (HCl) where positive Mn^+^ ions react with Cl^−^ to produce Cl_2_. An inert gas pushes Cl_2_ into another vessel containing potassium iodine (KI) where it reacts to produce I_2_. This mixture was titrated with sodium thiosulfate and the ratio of Mn^3+^ to Mn^4+^ was calculated by the stoichiometry of balanced chemical equations [16]. 

Electrical properties were found using the four-point technique in a cryogen-free superconducting setup. Four-point chips, measuring voltage and current separately were placed in applied magnetic fields of up to 5 T with a varied temperature between 10 K and 300 K. Resistance of the samples was recorded, resistivity was calculated using sample dimensions.

Magnetic measurements were made using a Vibrating Sample Magnetometer (VSM) in the temperature range of 4–300 K in external magnetic fields of up to 4 T.

## 3. Results

### 3.1. Structural Analysis

Visual inspection of stack XRD patterns, as shown in Figure 1, suggests possible decrease in cell dimensions due to the shift to the right of patterns with increasing Ca content. All samples are single phase with less than 5% levels of impurities. Wider peaks for nanocrystalline samples are indicative of smaller crystallites size in the compounds [17] which is further confirmed by Rietveld refinement analysis and Williamson–Hall method in Table 1 and Table 2.

Investigation of XRD data by Rietveld refinement analysis established orthorhombic (Pbnm) space group no. 62 for all samples, with decreasing cell dimensions and volume for each subsequent substitution of Sr. The results are illustrated in Figure 2. Stability of the structure in orthorhombic perovskite materials is assessed by the Goldschmidt tolerance factor. It was calculated using following relation [18,19]:(1)$$t=\frac{R_{A}+R_{0}}{\sqrt{2}\left(R_{B}+R_{0}\right)},$$
where R_A_ is the radius of A cation, R_B_ is the radius of B cation, and R_0_ is the radius of the anion. All samples fall within orthorhombic/rhombohedral tolerance range of 0.7–1 [20] (pp. 707–714).

An addition of smaller crystal radius Ca^2+^ (1.32 Å) atoms in place of Sr^2+^ (1.45 Å) into the structure increases disorder [8,19] and decreases average Mn–O bond length, as seen in Table 1 and Table 2, which plays a crucial role in “double exchange” interaction [8]. The average angle between Mn–O–Mn stays relatively the same throughout the range of substitutions at 157.72(3)°.

The Williamson–Hall (W-H) method is advantageous in that it takes into account strain between crystallites, as opposed to the Scherrer method, which does not [21]. Thus, results form W-H analysis are expected to be closer to the real values. A comparison between Rietveld refinement analysis and Williamson–Hall method for approximation of crystallite size can be performed by contrasting them with TEM and optical microscopy results. As can be observed in Table 1 for bulk samples, W-H and Rietveld results are similar, but are two orders of magnitude different from real values. This can be attributed to each grain containing many crystallites. In Table 2, for nano-sized samples, it can be seen that values from W-H calculations are on average larger than those performed by Rietveld analysis and closer to values from TEM. Real sizes for nanocrystalline particles are in the upper limit of nano dimensions at approximately 70–80 nm. Selected samples of optical microscope pictures are presented in Figure 3 and selected TEM pictures in Figure 4. 

### 3.2. Oxygen Content

Implementation of iodometric titration analysis for samples is a reliable method for determining deficiency or excess of oxygen in manganites [13,16]. All results are presented in Table 3. 

Titration of bulk compounds revealed oxygen deficiency for all samples. An average of O_2.93±0.02_ for x = 0.02, similar to other samples, shows larger than expected deficiency of approximately O_2.98_ [22]. This is attributed to preparation methods and difficulty in oxygenating the sample during calcination and sintering. Lower levels of oxygen would affect magnetic and electrical properties as it affects the stoichiometry and changes Mn^3+^/Mn^4+^ ratio which would lower the amount of “double exchange” interaction [13]. 

All nanocrystalline compounds exhibited small excess of oxygen stoichiometry. The level of excess did not exceed O_3.02±0.01_ as is with x = 0.02 sample. We suggest that the culprit for such result should be found in the high surface to volume ratio of nano-sized particle. Broken bonds on the surface will increase the ratio of Mn^4+^ to Mn^3+^ by attracting oxygen in the air. This, in turn, can create vacancies and affect its magnetic and electrical properties.

Finally, it can be observed that standard deviation, or deviation from the “mean”, is larger for bulk samples than the nano-sized samples. This is explained by the variation of oxygenation within the bulk. Nevertheless, all relative standard deviations do not exceed 3% and can be considered reliable.

### 3.3. Electrical Measurements

An investigation of electrical behavior for polycrystalline samples x = 0.02; 0.05; 0.1 and 0.2 was performed using the four-point probe method in external fields of μ_0_*H* = 0 T; 1 T; 2 T in temperature range of 10–290 K. For the sample with x = 0.3 fields of up to 5 T were used. Data were used to estimate metal–insulator transition temperature *T*_p_ and magnetoresistivity *MR* according to the following equation [23]:*MR*% = [(ρ(H) − ρ(0))/ρ(0)] × 100,(2)

The first observation to be made is that all samples, except x = 0.3, exhibit a transition temperature *T*_p1_ typically associated with grain boundary conditions [23,24,25]. Visual inspection shows that it is a wide and smooth transition, different from the usual sharp peaks associated with ferromagnetic–paramagnetic transition (and consequently metal-insulator) at *T*_p2_ (inside of the grains). The addition of Ca ions increases disorder, which tends to spread itself toward the edges of the grains in order to conserve energy [26,27]. Boundaries act as insulators or semi-conductors [27,28]. As can be seen in Figure 5, the application of external field shifts *T*_p_ to the right. This is caused by lowering spin fluctuations and delocalization of charge carriers [19].

A separation between *T*_p1_ and *T*_p2_ (which is associated with *T*_C_) is reported when grain boundary conditions are strong enough due to mismatch in ionic radius at the A-site [26]. Samples x = 0.02 and 0.05 do not show a peak *T*_p2_ because it is beyond the temperature range. Starting with x = 0.1 a sharp peak appears which lies higher in temperature than *T*_C_ value. With application of a field, *T*_p2_ tends to smooth out, as can be seen with the sample x = 0.2 in Figure 5b.

Special attention should be paid to the sample with x = 0.3. Its graph is presented in Figure 5c. Magnetic fields of up to 5 T were applied. For μ_0_H = 0 T; 1 T the sample conductivity is below the detection level for temperatures below 50 K. With the application of 2 T resistivity drops significantly. Its curve can be seen in the main portion of the figure. Curves for fields of 3–5 T can be seen in the inset and show further drop in resistivity and a shift to higher temperatures for the peak. This suggests that higher field overcomes intergrain resistance and further increase in the field induces a parallel alignment of manganese ions. In higher applied magnetic fields, the magnetic moments of the neighboring grains tend to be parallel, enhancing the tunneling of the conduction electrons similarly with a giant magnetoresistance effect (GMR) [26].

Magnetoresistivity is negative in all samples. Table 4 shows an increasing value of *MR* with increasing level of Ca substitution. The highest *MR*_Max_ (2 T) is 51.96% and 29.08% for 1 T for x = 0.2 at 217 K. Sample with x = 0.02 exhibits the lowest *MR*_Max_ of 11.9% and 4.45% for 2 T and 1 T at 289 K, respectively. A sample with x = 0.3 has *MR* of 77.62% between 3 T and 4 T at 118 K and 99.99% for 2–3 T at 130 K (shown in brackets in Table 4). In order to calculate MR, both the peak value of resistivity from the lower field data and an isothermal value from the higher field data are taken and then inserted into Equation (2). The highest peak resistivity ρ_peak_ of about 3 Ωcm at 275 K is observed in sample x = 0.05. The lowest ρ_peak_ for x = 0.3 at 5 T is 73 Ωcm (in brackets) at 174 K. All relevant data, including *T*_C_ and *T*_p_, are presented in Table 4.

### 3.4. Magnetic Properties

Investigation of magnetization vs. temperature (*M* vs. *T*), with the samples cooled in zero and in an applied magnetic field of μ_0_*H* = 0.05 T (ZFC-FC) was carried out in a vibrating sample magnetometer (VSM). All samples except x = 0.3 exhibit strong ferromagnetic behavior. Samples with x = 0.3 for both systems exhibit antiferromagnetic charge-exchange-type (CE-AFM) and charged ordered (CO) state. In CE-AFM structure, Mn^3+^ and Mn^4+^ are arranged like a checkerboard in the ab-plane (Mn-O_2_); exchange interaction causes Mn^3+^ e_g_ electron to occupy either d_3×2−r2_ or d_3y2−r2_ orbital. As a result, there are FM zig-zag chain arrangements which are AFM to each other and also stack antiferromagnetically in the c-plane. Figure 6 presents selected graphs of *M* vs. *T* with insets representing a derivative dM/dT of the plots for which the minimum is associated with *T*_C_. The addition of Ca^2+^ ions causes smaller cell dimensions; increased disorder and it lowers the Curie temperature [17]. The increase in Ca substitution lowers *T*_C_ further. Nano-sized particles show lower values of *T*_C_ than their bulk counterpart due to their size and surface effect, including broken bonds, canted spins, and reduced magnetization [29,30]. In addition, the curves for nano-sized samples are “smoother”, covering a wider temperature range in their ferromagnetic–paramagnetic change but have lower maximum value of magnetization *M*(*T*).

An observation of ZFC-FC curves, especially for bulk samples, reveals an upturn in magnetization at lower temperatures at around 70–100 K, an example of which is in Figure 6a. This can be attributed to the magnetization of praseodymium Pr ions [10]. Samples with x = 0.3 also exhibit increased magnetization at low temperatures starting at around 100 K, seen in Figure 6c,d. Maximum value is not as high as for the rest of the samples, approximately 0.04 μ_B_/f.u. for x = 0.3 vs. 1 μ_B_/f.u. for x = 0.1. Other CO compounds, such as La_0.4_Ca_0.6_Mn_0.9_Ga_0.1_O_3_ [11] and Pr_0.75_Na_0.25_MnO_3_ [14] have been reported with similar increase in magnetization due to suppression of CO state. The difference between La^3+^ and Pr^3+^ is their crystal ionic size (1.356 Å for La^3+^ and 1.319 Å for Pr^3+^ [20]) and their electron configuration: La^3+^ has no 4*f* electrons while Pr^3+^ has the configuration = [Xe]4*f*^2^. The size difference affects the bandwidth, with Pr-based manganites being referred to as narrow bandwidth manganites. Appearance of the charged ordered state is influenced by Mn^3+^/Mn^4+^ ratio as well as the bandwidth [31], therefore Pr based manganites enter CO state more easily. The low temperature increase in magnetization is associated with phase separation (PS), i.e., existence of FM clusters, forming among AFM matrix [11]. The graphs for x = 0.3 at higher temperatures show behavior typical of charged ordered state. A small feature at around 170 K represents Neél temperature, *T*_N_ (most visible in the bulk sample), and the peak at around 210 K representing the onset of the charge ordering phase, *T*_CO_. Such a behavior was revealed in several manganites systems such as La_0.250_Pr_0.375_Ca_0.375_MnO_3_ [32], Pr_0.7_Ca_0.3_MnO_3_ [33], La_0.3_Ca_0.7_Mn_0.8_Cr_0.2_O_3_ [34], Pr_0.57_Ca_0.41_Ba_0.02_MnO_3_ [35], Nd_0.5_Ca_0.5_MnO_3_ [36], and La_0.4_Ca_0.6_MnO_3_ [37], and it arises as a result of doping and/or due to the reducing of the sizes of the particles to the range of nanometers. 

The feature from 45 K, in bulk sample, could be the signature of the blocking of isolated spins between FM clusters as seen in spin glass materials [38,39].

According to Landau’s mean field theory [40], Gibbs free energy of the system around a critical point can be expanded in a Taylor series as:*G* (*T*,*M*) = *G*_0 + *MH* + *aM*^2^ + *bM*^4^ + …,(3)
where coefficients *a* and *b* depend on temperature. A derivative of the energy with respect to magnetization, to find the minima, results in an expression:*H*/*M* = 2*a* + 4*bM*^2^,(4)

An easy way to confirm the correctness of the mean field theory approach for given samples is the Arrott plot, i.e., *M*^2^ vs. *H*/*M* [41]. If isotherms are straight and parallel to each other around *T*_C_ then the assumption is correct. Observation of Arrott plots for our samples reveals curved, non-parallel lines for bulk compounds (Figure 7a,c) and almost straight lines for nano-sized compounds (Figure 7b,d). Furthermore, Banerjee criterion is a useful tool for determining the order of the phase transition [42]. According to the criterion, positive slope represents second order phase transition and negative slope corresponds to first order transition. Samples with x = 0, 0.02, 0.05, 0.1 show only positive slope. As can be seen in Figure 7c,d, samples with x = 0.2 show negative slope in the first stage of the magnetization at temperatures above *T*_C_ suggesting first order transition. Samples with x = 0.3, not pictured, show mostly negative slope plots. 

Inexactness of the exponents in the Equation (4) can be solved by constructing an Arrott–Noakes plot, i.e., *M*^1/β^ vs. μ_0_*H*/*M*^1/γ^ [43] with proper exponents β and γ. The exponent β relates to the spontaneous magnetization below *T*_C_ and γ relates to the inverse susceptibility above *T*_C_ [44] (pp. 7–10). An additional exponent δ relates magnetization and external field at *T*_C_. In mean field model: β = 0.5 and γ = 1, but there exist more models with different exponent values. The value of the exponents relates to the system dimensionality, spin, and range of the interaction *J*(r). In renormalization group theory [45], exchange interaction is defined as *J*(r) = 1/r*^d^*^+*σ*^ where *d*—dimensionality of the system and *σ*—range of interaction. For *σ* greater than 2: β = 0.355, γ = 1.366 and δ = 4.8—is the case of the 3D Heisenberg model. If *σ <* 3/2, long-range interactions occur, according to mean field theory. For the tricritical point, the critical exponents are: β = 0.25, γ = 1, and δ = 5 [46]. The tricritical point sets a boundary between two different ranges of order phase transitions (first order and second order). These exponents can be generalized into following equations [43]: *M*_S_(*T*) = *M*_0_ (−*ε*)*^β^*, *T* < *T*_C_,(5)
(6)$$\chi^{-1}\left(T\right)=\left(\frac{h_{0}}{M_{0}}\right)\varepsilon^{\gamma },\;T\;>\;T_{C},$$
*M = D* (μ_0_*H*^1*/δ*^), *T* = *T*_C_,(7)
where *ε* is the reduced temperature (*T* − *T*_C_)/*T*_C_ and *M*_0_, *h*_0_/*M*_0_, and *D* are critical amplitudes.

To find the proper exponents, we constructed Arrott–Noakes plots by implementing the Modified Arrott plot (MAP) method [43]. It is an iterative method. It includes, at first, construction of Arrott–Noakes plots by choosing exponents which produce straight parallel lines, with the line at *T*_C_ crossing at the origin. Secondly, we find intercepts of lines around *T*_C_; on the abscissa, above Curie temperature, to obtain the values of χ_0_^−1^ and on the ordinate, below *T*_C_, to find the values of *M*_s_. Finally, these values are plugged into the Equations (5) and (6) to obtain new values of β and γ. This process is repeated until results stabilize. The exponent δ is found using the Widom relation: β + γ = β δ [10]. Selected MAP graphs are shown in Figure 8. Full results for MAP are presented in Table 5.

Remarkably, polycrystalline samples are more closely governed by tricritical mean field model (β = 0.212, γ = 1.057, δ = 5.986 for x = 0.02), rather than 3D Heisenberg model as reported for other manganites in the literature [10,47]. Sample x = 0.3 does not exhibit ferromagnetic behavior in the high temperature range, only showing small magnetization at low temperatures due to FM clusters and Praseodymium ions, therefore, no critical values are presented in this work. Alternatively, all critical exponents for nano-sized particles fall within the mean field model values (β = 0.541, γ = 1.01, δ = 2.867 for x = 0.02) which is comparable to reported values in La_0.7_Ba_0.3−x_Ca_x_MnO_3_ compounds [22]. Similarly, to the bulk, critical values for nano-sized sample x = 0.3 are not presented. 

The Kouvel–Fisher (KF) method for determining critical exponents is a widely implemented tool for ferromagnetic materials [47,48,49]. We used KF to confirm the results from MAP as the two methods are regarded to be very accurate and reliable [47]. Akin to MAP, it is also an iterative method. It requires the construction of Arrott–Noakes plot, finding the intercepts on ordinate and abscissa and fitting these values in the equations [47,49]:*M*_s_ {d*M*_s_/d*T*}^−1^ = (*T* − *T*_C_)/β(8)
χ_0_^−1^ {d χ_0_^−1^/d*T*}^−1^ = (*T* − *T*_C_)/γ(9)

Ideally, plot of *M*_s_ {d*M*_s_/d*T*}^−1^ vs. *T* is a straight line with slope = 1/β and the intercept giving *T*_C_/ β. Same logic applies to the plot of χ_0_^−1^ {d χ_0_^−1^/d*T*}^−1^ vs. *T* which gives slope equal to 1/γ. We present, in Figure 9, selected examples of results from KF calculations compared to results from MAP for the same compounds. It is evident, that the lines in KF plots are close to being straight and their slope values result in β and γ being close to results from MAP. 

Magnetic entropy change was calculated using the following formula [50,51]:(10)$$\Delta S_{m}(T,H_{0})=S_{m}(T,H_{0})-S_{m}(T,0)=\frac{1}{\Delta T}\int_{0}^{H_{0}}\left[M(T+\Delta T,H)-M(T,H)\right]dH,$$
where magnetization *M*(μ_0_*H*) is taken from isothermal data at fields of 1–4 T. 

Plots of −Δ*S*_M_ vs. *T* (temperature) were constructed in order to better estimate magnetocaloric effect of the compounds, including calculations of relative cooling power RCP [51,52]:(11)$$RCP\left(S\right)=-\Delta S_{m}\left(T,H\right)\times \delta T_{FWHM}$$
where $\delta T_{FWHM}$ is the range of temperature at full width half maximum.

Selected graphs of samples’ entropy change are presented in Figure 10. It is noteworthy that, generally, polycrystalline samples exhibit higher maximum entropy change −Δ*S*_M_ compared to their nano-sized counterparts, as seen in Figure 10a vs. 10b. Bulk compound with x = 0.05 shows −Δ*S*_M_ (max) = 5.56 J/kgK at 4 T and nano-sized sample shows −Δ*S*_M_ (max) = 3.25 J/kgK at 4 T. Bulk samples with x = 0.1 and x = 0.2 exhibit high maximum entropy change at 6.9 J/kgK and 9.2 J/kgK respectively. For all samples, maximum entropy change occurs at around their Curie temperature *T*_C_. It is also important to note that nanocrystalline compounds exhibit wider range of effective temperature $\delta T_{FWHM}$ at around 40–70 K while bulk samples show range of 20–30 K. 

Construction of cooling equipment should not be “blindly” reliant on reported values of RCP [53]; nevertheless, for a long time it has been a useful tool in estimating the applicability of the magnetic material. Although it is the width of the temperature change in nano-sized compounds that is criticized as unreliable, in this work, nanocrystalline compounds deserve attention for their satisfactory level of entropy change, greater or close to 4 J/(kgK) for Δμ0H = 4 T. Both nano-sized and bulk systems exhibit values of entropy change and RCP comparable with other manganites reported in the literature [10,52,54,55].

Delving deeper, unfortunately, RCP tends to overestimate the merits of the materials which have a large temperature range of magnetocaloric effect (*δT_FWHM_*) but small entropy changes [55,56] in no small part because materials with same relative cooling power can behave differently in a magnetic cooling simulation [57]. Moreover, one of the last printed books about magnetic cooling ignores this figure of merit [58].

In order to be of use in magnetic refrigeration applications the chosen materials have to show a large magnetocaloric effect—a large magnetic entropy change. Besides this, there is a list of requirements related with heat transfer, heat capacity, heat conductivity, chemical and mechanical stability, hysteresis losses, eddy currents losses, etc. Therefore, magnetic measurements can be a trusted guide in deciding if a material deserves the effort of complete characterization. Some compounds of the type A_1−x_B_x_MnO_3_ can be considered to be of interest for magnetic refrigeration because some of these compounds show −Δ*S*_m_ values (−Δ*S*_m_ ≈ 4–6 J·kg^−1^·K^−1^ for Δ*H* = 5 T) comparable to intermetallic alloys [56]. This is the case of our investigated compounds.

The best way to decide the practicality of a material in magnetic cooling applications is by testing directly in a magnetic refrigerator, working on the principle of the AMRR (Active Magnetic Regenerative Refrigeration) cycle [12]. Recent investigations confirmed the magnetocaloric performance and the potential in magnetic refrigeration of the perovskite oxide Pr_0.65_Sr_0.35_MnO_3_ [12,59], which has the stoichiometry of our parent compound. When Ca substitutes for Sr in this compound, the magnetic parameters and magnetocaloric effect can be tuned for various temperature applications without significant changes in the magnitude of magnetic entropy change. Our investigations support the bulk polycrystalline Pr_0.65_Sr_(0.35−x)_Ca_x_MnO_3_ compounds to be promising for magnetic cooling technology, while lower magnitude of magnetocaloric effect in nanocrystalline samples, unfortunately, seems to somewhat hinder their direct practical applications [53,56,57].

Special mention should be made of some feature for samples with x = 0.3. Both bulk (Figure 10c) and nano-sized (not pictured) compounds exhibit positive entropy change at temperatures associated with antiferromagnetic (CO)/ferromagnetic/paramagnetic phase transition. This phase transition can be observed in the ZFC-FC plots in Figure 6 at around 200–210 K for both samples. As can be observed in Figure 10c, bulk sample exhibits inverse (materials cool down when a magnetic field is adiabatically applied) to normal MCE change suggesting AFM/FM transition, but it seems the changes in the lattice parameters can also modify the magnetic exchange interaction to give rise to a such behavior [60,61]. Additionally, Figure 10d shows a negative entropy change of 5.1 J/kgK at 4 T for nano-sized samples at 75 K. Similar behavior and even higher entropy change is exhibited by the bulk sample at around 40 K (not pictured) at the suppression of the AFM state. These materials can be used to produce both cooling and heating when they are adiabatically demagnetized. Unfortunately, the large values of the magnetic entropy changes can be reach only at low temperatures, far from the range of domestic refrigeration, in the case of these samples.

Low coercivity is of utmost importance in application of cooling materials [51]. All samples exhibit low coercive fields as measured in a hysteresis loop at external fields of up to 4 T at 4 K. Largest coercive field exhibited by a bulk sample is of 190 Oe for x = 0.2; with the smallest being 130 Oe for x = 0.05. Nanocrystalline samples carry larger coercive field compared to polycrystalline samples. It increases with decreasing size of the particles until single domain size. Particles become superparamagnetic with further decrease in size [62]. Largest coercive field is produced by the parent nano-sized with x = 0 at 810 Oe. All values for coercive field and magnetic saturation *M*_s_ are presented in Table 6 and Table 7. Nanocrystalline sample x = 0.3 does not reach saturation at 4 T.

## 4. Conclusions

Two sample systems of Pr_0.65_Sr_0.35−x_Ca_x_MnO_3_ (x = 0.02, 0.05, 0.1, 0.2, 0.3) were prepared. Solid state reaction as a method of preparation of polycrystalline compounds resulted in samples with an average 12,000 nm grain size. Sucrose based sol–gel method for production of nanocrystalline samples resulted in particles of an average size of approximately 70 nm. X-ray diffraction measurements revealed a single crystallographic phase for all samples. Rietveld refinement analysis confirmed single phase, orthorhombic (Pbnm) symmetry, diminishing cell volume and Mn–O bond length with increasing Ca substitution. Iodometric titration was implemented on both systems to determine their oxygen content. All bulk samples show oxygen deficiency close to O_2.93±0.02_ attributed to the preparation method. All nano-sized samples exhibit small oxygen excess of O_3.02±0.01_ or less, credited to the effects of the surface/volume ratio of the particles. Bulk compounds were measured for their electrical properties and revealed a feature at a temperature *T*_p1_ associated with high disorder at grain boundaries due to Ca substitution. Additionally, samples with Ca levels x = 0.1, 0.2 exhibit a peak at a temperature *T*_p2_ associated with the ferromagnetic-paramagnetic transition. With the increase in applied magnetic field, *T*_p2_ tends to become higher and smooth out. The sample with x = 0.3 possesses “infinite” resistance for μ_0_*H* = 0 T and 1 T at around 50 K. Further increase in applied field up to 5 T results in overcoming intergrain resistance and shift in the resistance peak. Negative magnetoresistivity was observed for all bulk samples. The smallest value of *MR* was 11.99% for x = 0.02 at 2 T at *T*_p1_, while the largest value of *MR* for x = 0.2 of 51.96% for *T*_p2_. Zero-field cooled–field cooled graphs reveal ferromagnetic behavior for x = 0.02, 0.05, 0.01, 0.2 samples. Both x = 0.3 show ferromagnetic-like behavior at low temperatures and antiferromagnetic behavior at higher temperatures than 80 K, with transition to ferromagnetic to paramagnetic behavior at 210 K. Curie temperature *T*_C_ lowers with each Ca substitution: 273 K for bulk x = 0.02 compared to 295 K for parent compound; 252 K for nano-sized x = 0.02 vs. 257 K for parent compound. Low coercivity was found for all samples. Nanocrystalline samples exhibiting larger coercivity compared to their bulk counterparts: 720 Oe vs. 180 Oe for x = 0.02. Arrott plots confirm second order phase transition for all samples except for x = 0.3. Modified Arrott plot (MAP) analysis disclosed critical behavior. Critical exponents for polycrystalline samples belong to tricritical mean field model while exponents for nano-sized particles belong to mean field model. The Kouvel–Fisher (KF) method for analyzing critical exponents confirmed MAP results. Bulk compounds reveal higher magnetic saturation *M*_s_ and maximum entropy change Δ*S*_M_ than nano-sized compounds. Bulk sample with x = 0.2 exhibits highest entropy change Δ*S*_M_ = 9.21 J/kgK (4 T) at 204 K. Relative cooling powers (RCP) for equivalent bulk and nano-sized compounds are comparable in value: 166 J/kg vs. 175 J/kg (4 T) for bulk and nano-sized with x = 0.02 This is due to the wide effective temperature range of entropy change *δT_FWHM_* in nanocrystalline samples. All values of RCP are high and comparable with relevant compounds reported in the literature. Bulk samples with x = 0.3 exhibits positive entropy change at antiferromagnetic-ferromagnetic transition temperature Magnetic entropy changes and magnetic parameters of the bulk polycrystalline samples indicate them as potential candidates for magnetocaloric materials. Possibly, they can be combined in construction of multistep refrigeration processes to increase their temperature range and effectiveness. The nanocrystalline samples, which show comparatively lower maximum entropy change are moved to the immediate background of the picture of usable magnetocaloric compounds but could, possibly, be still useful in future developments.

## Figures and Tables

**Figure 1 nanomaterials-13-01373-f001:**
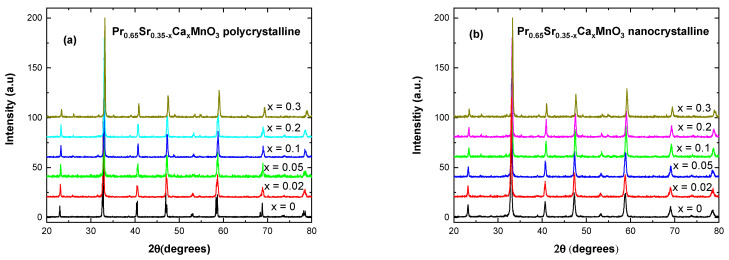
X-ray diffraction patterns for (**a**) Pr_0.65_Sr_0.35−x_Ca_x_MnO_3_ polycrystalline bulk samples and (**b**) Pr_0.65_Sr_0.35−x_Ca_x_MnO_3_ nano-sized samples.

**Figure 2 nanomaterials-13-01373-f002:**
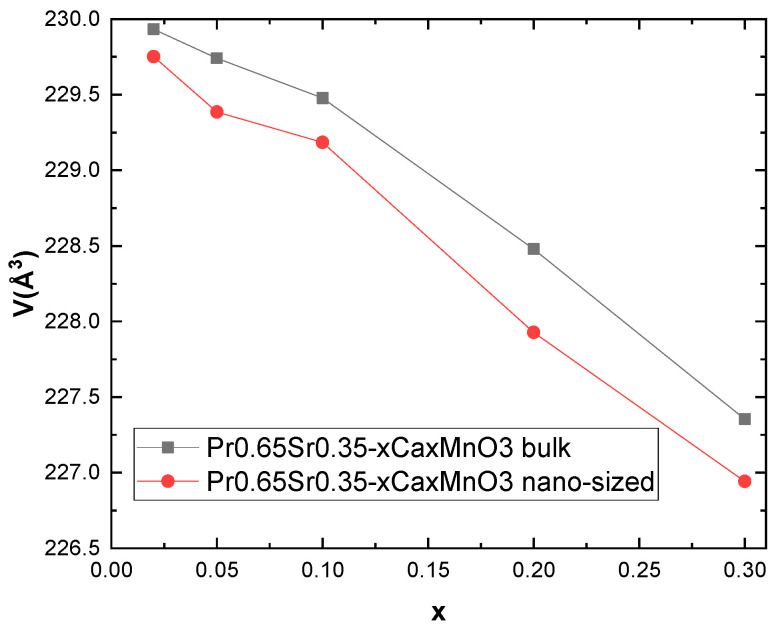
Plot of cell volume change for polycrystalline and nanocrystalline samples.

**Figure 3 nanomaterials-13-01373-f003:**
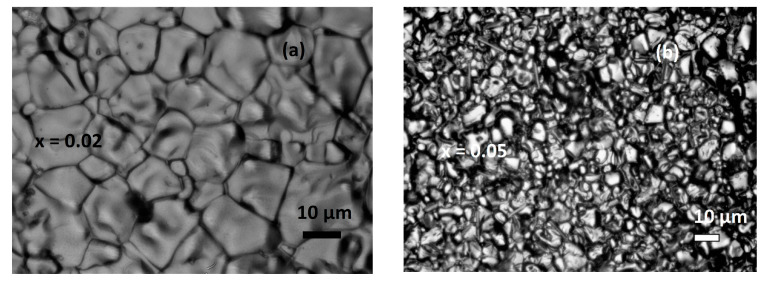
Selected optical microscope pictures for Pr_0.7_Sr_0.3−x_Ca_x_MnO_3_ bulk samples (**a**) for x = 0.02, (**b**) x = 0.05.

**Figure 4 nanomaterials-13-01373-f004:**
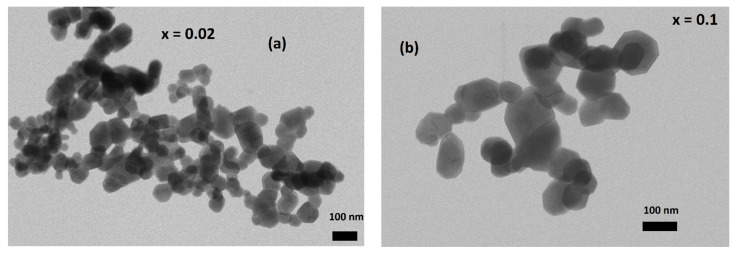
Selected TEM pictures for Pr_0.7_Sr_0.3−x_Ca_x_MnO_3_ nano-sized samples for x = 0.02 (**a**), x = 0.01 (**b**).

**Figure 5 nanomaterials-13-01373-f005:**
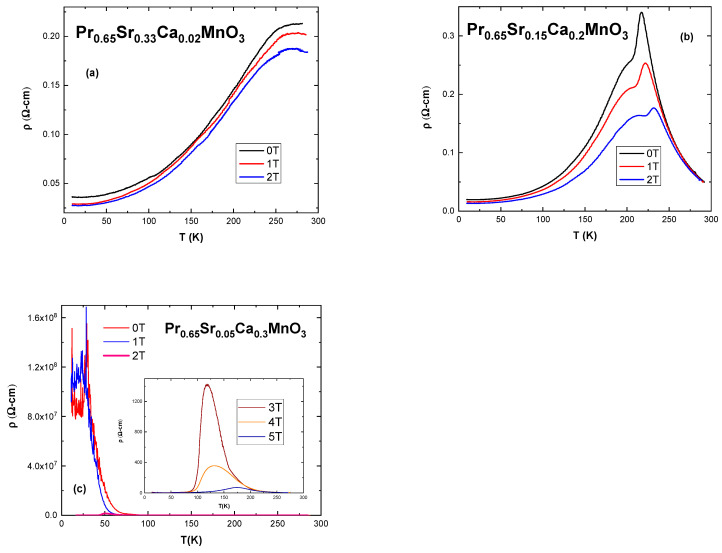
Graphs of resistivity vs. temperature for bulk samples: (**a**) x = 0.02, (**b**) x = 0.2, (**c**) x = 0.3; The inset shows plot of resistivity at higher fields which are not visible in the main plot.

**Figure 6 nanomaterials-13-01373-f006:**
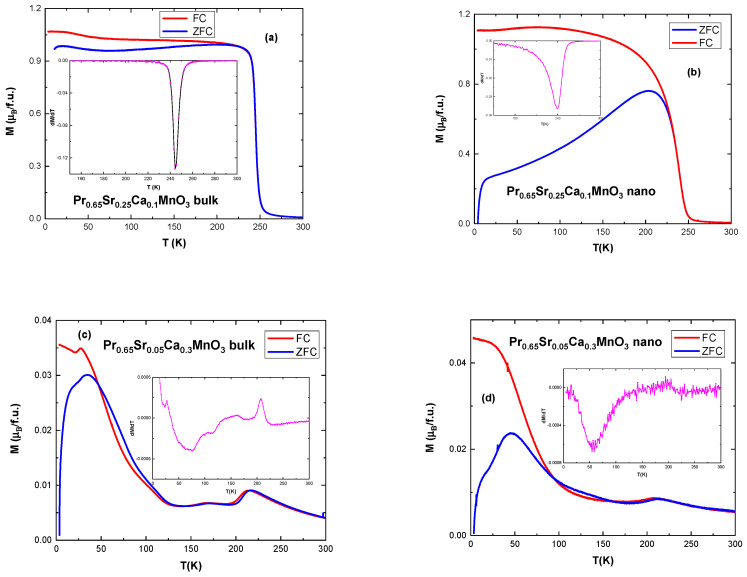
ZFC-FC plots in external field of 0.05 T (**a**) ZFC-FC curves and derivative of magnetization (inset) for the bulk sample Pr_0.65_Sr_0.25_Ca_0.1_MnO_3_; (**b**) ZFC-FC curves and derivative (inset) for nanocrystalline sample Pr_0.65_Sr_0.25_Ca_0.1_MnO_3_. (**c**) ZFC-FC curves and derivative shown as inset, for polycrystalline sample Pr_0.65_Sr_0.05_Ca_0.3_MnO_3_; (**d**) ZFC-FC curves and derivative shown in the inset for nanocrystalline sample Pr_0.65_Sr_0.05_Ca_0.3_MnO_3_.

**Figure 7 nanomaterials-13-01373-f007:**
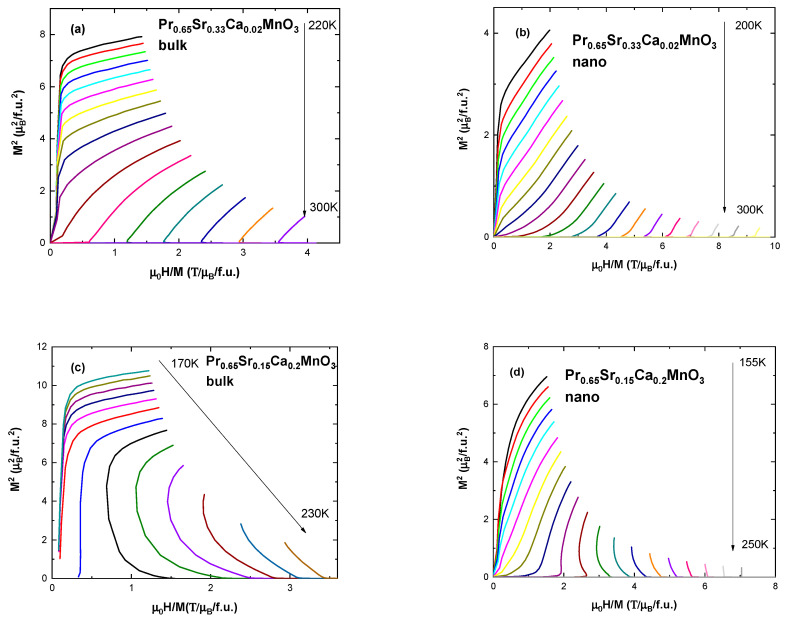
Arrott plot (*M*^2^ vs. *H/M*) for (**a**) the bulk sample with x = 0.02; (**b**) the nanocrystalline sample with x = 0.02; (**c**) bulk sample with x = 0.2; (**d**) nanocrystalline sample x = 0.2.

**Figure 8 nanomaterials-13-01373-f008:**
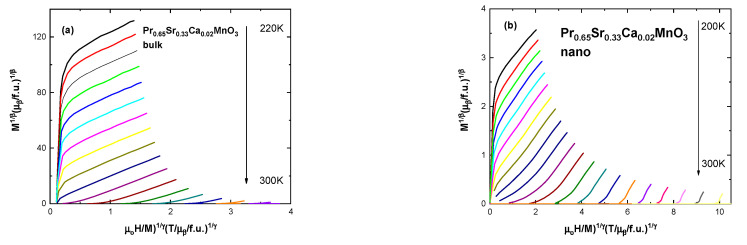
Modified Arrott plots for (**a**) the bulk sample x = 0.02 with β = 0.212, γ = 1.057, δ = 5.986 and for (**b**) the nanocrystalline sample x = 0.02 with β = 0.541, γ = 1.01, δ = 2.867.

**Figure 9 nanomaterials-13-01373-f009:**
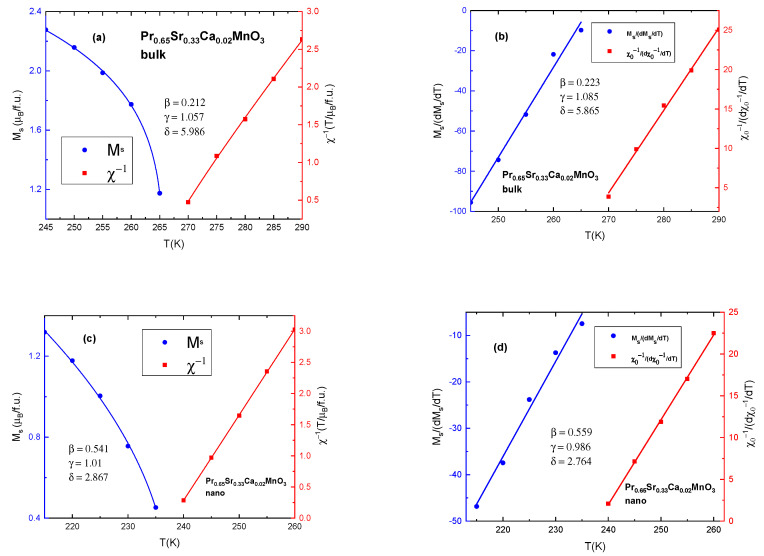
Calculated values for critical exponents for (**a**) MAP analysis for the bulk sample with x = 0.02 (**b**) KF analysis for the bulk sample with x = 0.02. (**c**) MAP analysis for the nanocrystalline sample x = 0.02 (**d**) KF analysis for nanocrystalline sample x = 0.02.

**Figure 10 nanomaterials-13-01373-f010:**
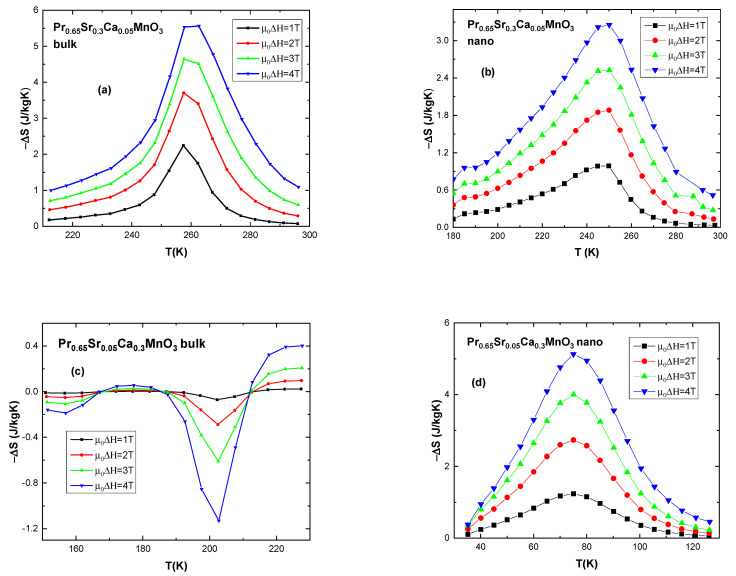
Magnetic entropy change vs. temperature for selected samples: (**a**) x = 0.05 bulk; (**b**) x = 0.05 nanocrystalline sample; (**c**) x = 0.3 bulk at AFM-FM; (**d**) x = 0.3 nanocrystalline at low temperature.

**Table 1 nanomaterials-13-01373-t001:** Calculated tolerance factors, Mn-O lengths, and crystallite sizes for polycrystalline samples using the Williamson–Hall and Rietveld methods and including particle diameters by optical microscopy.

Ca Content (Bulk)	*t* (Tolerance Factor)	Mn-O (Å)	Mn-O-Mn	Average Particle Diameter (μm)	Williamson–Hall Size (nm)	Average Rietveld Size (nm)	Strain
			(°)				
x = 0.02	0.925	1.964	157.73	15	111.33	96.67	0.0022
x = 0.05	0.924	1.962	157.74	11	156.63	113.68	0.0019
x = 0.1	0.921	1.961	157.73	13	132.88	192.59	0.0024
x = 0.2	0.917	1.956	157.69	10	144.85	125.64	0.0023
x = 0.3	0.912	1.954	157.72	11	123.56	86.79	0.0021

**Table 2 nanomaterials-13-01373-t002:** Calculated Mn-O lengths and crystallite sizes for nanocrystalline samples using the Williamson–Hall and Rietveld methods and including grain diameters from TEM.

Ca Content (Nano)	Mn-O (Å)	Mn-O-Mn	Average Particle Diameter (nm)	Williamson–Hall Size (nm)	Average Rietveld Size (nm)	Strain
		(°)				
x = 0	1.972	157.74	68.1	72.45	54.33	0.0018
x = 0.02	1.964	157.75	64.8	71.15	45.64	0.0019
x = 0.05	1.958	157.69	78.2	84.59	57.21	0.0016
x = 0.1	1.957	157.69	87.5	82.95	39.75	0.0019
x = 0.2	1.955	157.7	71.5	66.73	45.62	0.0017
x = 0.3	1.951	157.71	65.7	69.69	51.11	0.002

**Table 3 nanomaterials-13-01373-t003:** Average oxygen content calculated using iodometry for bulk and nanocrystalline samples.

Ca Content	Average Mn^3+^/Mn^4+^ Ratio	Standard Deviation	Relative Standard Deviation (%)	Average Oxygen Content
x = 0.02 bulk	0.788	0.0167	2.12	O_2.93±0.02_
x = 0.05 bulk	0.783	0.0121	1.55	O_2.93±0.01_
x = 0.1 bulk	0.778	0.0139	1.79	O_2.94±0.02_
x = 0.2 bulk	0.781	0.0182	2.33	O_2.94±0.02_
x = 0.3 bulk	0.765	0.0226	2.95	O_2.94±0.02_
x = 0 nano	0.624	0.008	1.28	O_3.01±0.01_
x = 0.02 nano	0.612	0.009	1.47	O_3.02±0.01_
x = 0.05 nano	0.622	0.011	1.33	O_3.01±0.01_
x = 0.1 nano	0.615	0.009	1.77	O_3.02±0.01_
x = 0.2 nano	0.633	0.008	1.26	O_3.01±0.01_
x = 0.3 nano	0.638	0.012	1.88	O_3.01±0.01_

**Table 4 nanomaterials-13-01373-t004:** Experimental values for Pr_0.65_Sr_0.35−x_Ca_x_MnO_3_ bulk materials: electrical properties.

Compound (Bulk)	*T*_C_ (K)	*T*_p1_ (K) (*T*_p2_ (K))	ρ_peak_ (Ωcm) in 0 T	*MR*_Max_ (%) (1 T)	*MR*_Max_ (%) (2 T)
Pr_0.65_Sr_0.33_Ca_0.02_MnO_3_	273	274	0.213	4.45	11.99
Pr_0.65_Sr_0.3_Ca_0.05_MnO_3_	261	273	3.094	12.28	22.75
Pr_0.65_Sr_0.25_Ca_0.1_MnO_3_	244	256 (258)	1.602	23.28	33.27
Pr_0.65_Sr_0.15_Ca_0.2_MnO_3_	201	210 (217)	0.341	29.08	51.96
Pr_0.65_Sr_0.05_Ca_0.3_MnO_3_		-	>100 × 10^8^ (72.985 in 5 T)	77.62 (between 3 and 4 T)	99.99 (between 2 and 3 T)

**Table 5 nanomaterials-13-01373-t005:** Critical exponent values for all samples from modified Arott plot method.

Compound		γ	β	δ	*T*_C_ (K)
x = 0.02	bulk	1.057	0.212	5.986	273
x = 0.05	bulk	0.981	0.232	5.228	261
x = 0.1	bulk	0.985	0.25	4.94	244
x = 0.2	bulk	1.036	0.217	5.774	201
x = 0.3	bulk	-	-	-	-
x = 0	nano	1.023	0.552	2.853	257
x= 0.02	nano	1.01	0.541	2.867	252
x = 0.05	nano	0.986	0.508	2.941	249
x = 0.1	nano	0.977	0.512	2.908	239
x = 0.2	nano	0.967	0.531	2.821	191
x = 0.3	nano	-	-	-	-
Mean field model		1	0.5	3	
3D Heisenberg model		1.366	0.355	4.8	
Ising model		1.24	0.325	4.82	
Tricritical mean field model		1	0.25	5	

**Table 6 nanomaterials-13-01373-t006:** Experimental values for Pr_0.65_Sr_0.35−x_Ca_x_MnO_3_ bulk materials: magnetic measurements.

Compound (Bulk)	*T*_C_ (K)	*M*_s_ (μ_B_/f.u.)	*H*_ci_ (Oe)	|Δ*S*_M_| (J/kgK) μ_0_Δ*H* = 1 T	|Δ*S*_M_| (J/kgK) μ_0_Δ*H* = 4 T	RCP (*S*) (J/kg) μ_0_Δ*H* = 1 T	RCP (*S*) (J/kg) μ_0_Δ*H* = 4 T	Refs.
Pr_0.65_Sr_0.35_MnO_3_	295			2.3				[12]
Pr_0.65_Sr_0.33_Ca_0.02_MnO_3_	273	3.71	180	2.04	5.53	34	166	This work
Pr_0.65_Sr_0.3_Ca_0.05_MnO_3_	261	3.78	170	2.24	5.56	44	167	This work
Pr_0.65_Sr_0.25_Ca_0.1_MnO_3_	244	3.94	190	3.03	6.91	45	186	This work
Pr_0.65_Sr_0.15_Ca_0.2_MnO_3_	201	3.97	160	4.48	9.21	60	270	This work
Pr_0.65_Sr_0.05_Ca_0.3_MnO_3_		3.81	170	4.6 (40 K)	15.3 (40 K)	92 (40 K)	380 (40 K)	This work
La_0.7_Ca_0.3_MnO_3_	256			1.38		41		[10]
La_0.7_Sr_0.3_MnO_3_	365			-	4.44 (5 T)		128 (5 T)	[10]
La_0.6_Nd_0.1_Ca_0.3_MnO_3_	233			1.95		37		[10]
Gd_5_Si_2_Ge_2_	276			-	18 (5 T)	-	535 (5 T)	[10]
Gd	293			2.8		35		[10]

**Table 7 nanomaterials-13-01373-t007:** Experimental values for Pr_0.65_Sr_0.35−x_Ca_x_MnO_3_ nano materials.

Compound (Nano)	*T*c (K)	*M*_s_ (μ_B_/f.u.)	*H*_ci_ (Oe)	|Δ*S*_M_| (J/kgK) μ_0_Δ*H* = 1 T	|Δ*S*_M_| (J/kgK) μ_0_Δ*H* = 4 T	RCP(*S*) (J/kg) μ_0_Δ*H* = 1 T	RCP(*S*) (J/kg) μ_0_Δ*H* = 4 T	Refs.
Pr_0.65_Sr_0.35_MnO_3_	257	3.6	810	1.62	4.55	54	263	This work
Pr_0.65_Sr_0.33_Ca_0.02_MnO_3_	252	3.08	720	0.69	2.5	38	175	This work
Pr_0.65_Sr_0.3_Ca_0.05_MnO_3_	249	3.16	540	0.99	3.25	48	178	This work
Pr_0.65_Sr_0.25_Ca_0.1_MnO_3_	239	3.49	510	1.41	4.37	39	215	This work
Pr_0.65_Sr_0.15_Ca_0.2_MnO_3_	191	3.39	620	1.08	3.9	41	185	This work
Pr_0.65_Sr_0.05_Ca_0.3_MnO_3_		-	600	1.23(75 K)	5.12(75 K)	48(75 K)	204(75 K)	This work
La_0.67_Ca_0.33_MnO_3_	260				0.97 (5 T)		27 (5 T)	[63]
Pr_0.65_(Ca_0.6_Sr_0.4_)_0.35_MnO_3_	220			0.75		21.8		[64]
La_0.6_Sr_0.4_MnO_3_	365			1.5		66		[17]

## Data Availability

Data presented in this study are available in this article.

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
