# Peer review of "Magnetic Properties and Magnetocaloric Effect of Polycrystalline and Nano-Manganites Pr0.65Sr(0.35−x)CaxMnO3 (x ≤ 0.3)"

_nanomaterials, 2023, doi:10.3390/nano13081373_

Round 1

Reviewer 1 Report

Since a Rietveld analysis has been performed, more data on crystallography of the materials is available and must be included.

A comment on charge compensation due to the change of stoichiometry could be included, for example are vacancies possible or any other mechanism?  

Some errors in the references section, please recheck.  

Reviewer 2 Report

The studies of near room temperature magnetocaloric materials ore particularly interesting. The paper reports a quite comprehensive and versatile experimental study of magnetic and magnetocaloric properties of bulk, polycrystalline and nanocrystalline Pr_0.65Sr_(0.35-x)Ca_xMnO_3 compounds. The synthesis of the systems in question is carefully described. Then the structural analysis is reported, based on XDR and TEM. It is followed by iodometry results and resistivity measurements. Then the usual magnetometric studies are discussed, with detailed insight into the critical behaviour of the samples based on the modified Arrot plot and Kouvel-Fisher approach as well as determination of the isothermal entropy change and RCP. The whole discussion is quite well-written and convincing and the results are of interest and use to the community of nanoscience. Before I recommend the manuscript for publication in Nanomaterials journal, I would recommend the Authors to address the detailed minor points listed below (mainly focused on presentation clarity):

Line 15” “Pbnm space group” should sound better than “Pbnm symmetry”.

Line 25: The term “effective entropy change temperature” does not sound clear enough (is that deltaT_FWHM?).

Line 99: Usage of “Williamson-Hall (W-H)” here would introduce the abbreviation W-H earlier than in line 145.

Line 125: Maybe the term “space group no. 62” would be added to Pbnm.

Table 1: Average particle size would be expressed in mm, not nm for clarity (the large number of digits would suggest high accuracy).

Table 3: For full clarity, “Average Mn3+ Ratio” would be replaced with “Average Mn3+/Mn4+ Ratio”.

Line 189: “estimate metal – insulator transition temperature” would sound better.

Line 211/212: The phrase “the sample exhibits “infinite” resistivity” might be replaced with “the sample conductivity is below the detection level” or similar statement.

Line 214: “insert”->”inset”.

Table 4: The data for Pr_0.65Sr_0.05Ca_0.3MnO_3  would be: resistivity rho_peak 10^10 (is that a measured value or “at least” this value? If yes, >10^10 would be better). Also, “72.985 in 5T” would be more clear. Also, “between 3 T and 4 T” etc. in the same row (for MR) might be better.

Line 237: What is CE type?

Line 279, Eq. 3: G_0 instead of Go.

Line 305” “is defined as” would be better.

Figure 9:

* in panels b and d it is not necessary to connect the points with straight lines; it is enough to show the data points and the least-squares fit of a linear function.

* in panels a and c, instead of joining the points with straight lines, a fitted power-law smooth curve should be shown.

Line 357, Eq. 11: The quantity deltaT_FWHM should be defined in the text.

Line 443: “Kouvel-Fisher”.

Line 546: The journal title should be written in italics.

Reviewer 3 Report

The manuscript is devoted to the investigations of bulk and nano-sized Pr0.65Sr(0.35-x)CaxMnO3 compounds in the vicinity of the Curie point. The results are interesting and should be published. However, the referee has examined this manuscript and regrettably concludes that it could be published only after minor revisions. The authors of this study conclude that "Both systems are a worthy addition to the growing list of perovskite manganites applicable in magnetic refrigeration" due to their high RCP values. I disagree with the authors. RCP is now known to be a poor description of a good magnetocaloric material, please see [K.G. Sandeman, Scripta Materialia 67 (2012) 566–571], [Anders Smith , Adv. Energy Mater. 20122, 1288–1318]. More than that, in the last printed book devoted to magnetocaloric refrigerators, ‘Magnetocaloric Energy Conversion: From Theory to Applications ‘ by Andrej Kitanovski, the author completely ignores this quantity. Since the majority of the recently created prototypes are based on the Active Magnetic Regenerator (AMR) cycle, which provides the highest achieved temperature spun, it should be mentioned that the magnetocaloric material is more suitable for an application in the AMR if it has a greater adiabatic temperature change on account of the smaller isothermal entropy change. This is strongly related to the heat transfer between the material and the heat transfer medium, since the heat transfer irreversibility losses can strongly reduce the device’s performance in the case of a small adiabatic temperature change. The materials obtained by the authors in this article provide us with low isothermal magnetic entropy change and can be recognized as promising for magnetic cooling technology. I advise to exclude the comparative analysis based on the RCP criteria. 
